# "Seeing Is Believing"—In-Depth Analysis by Co-Imaging of Periodically-Poled X-Cut Lithium Niobate Thin Films

**Sven Reitzig** [1,*], **Michael Rüsing** [1], **Jie Zhao** [2,†], **Benjamin Kirbus** [1], **Shayan Mookherjea** [2] **and Lukas M. Eng** [1,3]

1 Institut für Angewandte Physik, Technische Universität Dresden, 01062 Dresden, Germany; Michael.Ruesing@tu-dresden.de (M.R.); benjamin.kirbus@tu-dresden.de (B.K.); lukas.eng@tu-dresden.de (L.M.E.)
2 Department of Electrical and Computer Engineering, University of California, San Diego, CA 92161, USA; jie.2.zhao@nokia-bell-labs.com (J.Z.); smookherjea@ucsd.edu (S.M.)
3 ct.qmat: Dresden-Würzburg Cluster of Excellence—EXC 2147, TU Dresden, 01062 Dresden, Germany
* Correspondence: sven.reitzig@tu-dresden.de; Tel.: +49-351-463-43354
† Since February 2021 affiliated with Nokia Bell Labs, Murray Hill, NJ 07974, USA.

**Abstract:** Nonlinear and quantum optical devices based on periodically-poled thin film lithium niobate (PP-TFLN) have gained considerable interest lately, due to their significantly improved performance as compared to their bulk counterparts. Nevertheless, performance parameters such as conversion efficiency, minimum pump power, and spectral bandwidth strongly depend on the quality of the domain structure in these PP-TFLN samples, e.g., their homogeneity and duty cycle, as well as on the overlap and penetration depth of domains with the waveguide mode. Hence, in order to propose improved fabrication protocols, a profound quality control of domain structures is needed that allows quantifying and thoroughly analyzing these parameters. In this paper, we propose to combine a set of nanometer-to-micrometer-scale imaging techniques, i.e., piezoresponse force microscopy (PFM), second-harmonic generation (SHG), and Raman spectroscopy (RS), to access the relevant and crucial sample properties through cross-correlating these methods. Based on our findings, we designate SHG to be the best-suited standard imaging technique for this purpose, in particular when investigating the domain poling process in x-cut TFLNs. While PFM is excellently recommended for near-surface high-resolution imaging, RS provides thorough insights into stress and/or defect distributions, as associated with these domain structures. In this context, our work here indicates unexpectedly large signs for internal fields occurring in x-cut PP-TFLNs that are substantially larger as compared to previous observations in bulk LN.

**Keywords:** thin film lithium niobate; TFLN; LNOI; x-cut LN; ferroelectric domains; domain walls; piezoresponse force microscopy; second-harmonic generation; Raman scattering



## 1. Introduction

In recent years, periodically-poled thin film lithium niobate (PP-TFLN) has emerged as a promising platform for realizing modern-type integrated nonlinear and quantum optical devices. When compared to the standard bulk periodically-poled lithium niobate (PPLN) platform, PP-TFLN offers higher conversion efficiencies, a significantly reduced footprint size, as well as an excellent integrability into nanooptical systems [1–7]. Unlike bulk PPLN, the application of the TFLN platform is not limited by the restrictions that weak optical confinement imposes to footprint and device integration [8–10]. Subwavelength optical confinement is provided by a large change of the refractive index between TFLN and the substrate material. This is mandatory and fundamental in order to achieve an ultra-high frequency conversion for next-generation applications.

Similar to other ferroelectrics, efficient nonlinear conversion may be achieved by applying the straightforward concept of quasi phase matching (QPM) to periodically-poled

ferroelectric (FE) domain structures. For efficient and narrow-band QPM devices, it is thus required that the domain grid matches closely with its design parameters [11,12]. In particular, early studies on nonlinear optical devices in TFLN reported on strong deviations between simulated and effectively measured conversion efficiencies and spectra, a fact that was attributed to inhomogeneities in the poling procedure. Nevertheless, these assertions were off ground, and could not be verified to date, since they simply lack the appropriate analysis methods [1]. Hence, adequate inspection techniques are needed to quantify the necessary parameters, to the benefit of proposing dedicated process protocols for the optimized and highly accurate domain fabrication in TFLN. Key indicators for high-grade domain grids in TFLN are: (a) The appropriate poling period; (b) the duty cycle and homogeneity; (c) the domain grid length; and (d) the depth of poled domains. Therefore, inspection methods need to provide access to these parameters.

**(a) Poling period inspection (resolution):** Due to the strong confinement and the associated large dispersion of optical modes, TFLN requires poling periods that are significantly shorter as compared to bulk LN. For a typical application at telecom frequencies, poling periods $\Lambda$ between 2 and 5 µm must be realizable in TFLN, while for similar devices in bulk LN 10–30 µm periods usually would be sufficient [2]. This makes the fabrication of domain grids in TFLN much more demanding. Furthermore, recent works have demonstrated that even sub-µm periodicities in PP-TFLN waveguides can be manufactured [13,14], an achievement that has not yet been possible with bulk devices. Therefore, any investigation method must provide an optical resolution in the lower µm and even sub-µm range, as well as be able for imaging over the full length of the poled areas.

**(b) Duty cycle and homogeneity control (imaging contrast):** For an optimal conversion efficiency, a duty cycle of the periodic domain grid close to 50% is desired. Furthermore, irregular variations in the duty cycle will significantly broaden the conversion spectrum and hence reduce that efficiency [11,12]. Therefore, methods with a clear imaging contrast are required to analyze the duty cycle along its full poling length.

**(c) Domain grid length (speed):** The nonlinear conversion efficiency scales quadratically with the interaction length, while the spectral width decreases. Hence, for narrow band, low pump power, and high efficiencies, device lengths in the range of mm to cm or even longer are requested [2,9,15]. Ideally, an appropriate inspection method then must be able to investigate these large areas in a reasonable time, i.e., within less than some hours, solely to facilitate fabrication and structure optimization.

**(d) In-depth domain growth control (depth sensitivity):** To achieve optimal efficiencies, all inverted domains along the waveguide structure need to penetrate across the full depth of the TFLN. In most designs, x-cut TFLN is used for fabrication, facilitating domain poling via surface electrodes that are easily fabricated and removed by standard lithography, deposition, and etching techniques. Hence, the PP domain structure is arranged in-plane, requiring that domain growth fully proceeds along both the in-plane polar axis and into the TFLN (non-polar) depth, as shown in Figure 1a. To optimize the fabrication of PP domain grids as well as for delivering decent conversion estimates, these depths of inverted domains need to be known and measured with high accuracy. Typical optical applications involve TFLN feature thicknesses of 300–800 nm. Hence, imaging methods that offer a depth-sensitivity of a few nm are highly desirable.

Furthermore, it is highly beneficial that an imaging method is not only sensitive to the domain structure, but may be able to simultaneously detect parameters that potentially might influence the domain growth dynamics and/or the linear and nonlinear optical properties [16]. Examples into this context are defects, dopant concentration variations (e.g., from waveguide diffusion [17]), built-in electric fields or inhomogeneous stress distributions [18]. Here, an appropriate imaging method might allow analyzing the interaction between the domain structures and its growth, as well as other influencing factors. It is known from many previous works that domain walls (DWs) are accompanied by defects, electric fields, and significant mechanical stress, that might reach out by as much as some µm [19]. For TFLN, this is particularly relevant, since, on the one hand, typical film

thicknesses are on the order of 500 nm, resulting in extremely strained domain structures. On the other hand, poling periods in TFLN are significantly shorter than in bulk LN, which might impact the material properties or lead to DW-DW-interactions [20]. Additionally in x- or y-cut TFLN, other sources such as strong built-in fields stemming from charged head-to-head (h2h) and tail-to-tail (t2t) DWs need to be mentioned. Hence, a fundamental study of this broad palette of influencing parameters especially in TFLN is highly desired, since having been mostly neglected so far.

In total, it is critical to apply imaging techniques to the domain structures that enable monitoring and quantifying the aforementioned key performance indicators, in order to optimize the conversion efficiencies of PP-TFLN based devices. Several analysis methods have been applied so far, with limited success:

In situ transmission monitoring methods [4,21] are very fast and non-destructive, for example, as electrooptical [21] or SH-conversion-efficiency [4] monitoring of that amount of light which is transmitted through the waveguide while poling the surrounding. The poling procedure is then stopped when reaching a preset indicator value, e.g., a certain peak efficiency. While these methods provide almost instant feedback, they only deliver averaged and indirect information on the duty cycle and poling quality. In addition, they provide no insights into the microscopic domain distribution, the shape, depth or additional influence factors as mechanical stress.

For z-cut PPLN bulk wafers, selective etching can be considered as "the" standard technique. Selective etching [1] relies on the fact that oppositely-poled crystalline facets along the FE axis of LN show different etching rates in an hydrofluoric acid (HF) etch solution, typically exhibiting a deeper etch on the negatively-poled face [22]. This allows directly accessing the domain shape via topography on the z-surface, which is then analyzed by standard techniques such as light microscopy, AFM techniques, profilometers or electron microscopy. While being widely applied for z-cut bulk devices due to its large-scale imaging potential, it might be readily applicable to z-cut TFLN structures, as well [6]. Nevertheless, its destructive nature prohibits an application for standard process control in thin films: A conservative estimate of the etch-depth resolution certainly yields values on the order of 10–100 nm. While for a bulk crystal with a waveguide depth in the order of several µm, a surface grid of a few tens of nanometer can be polished without impacting the design properties of a waveguide structure. Nevertheless, this is not possible in TFLN. Here, even a thickness change on the order of tens of nanometers will significantly impact the propagating modes [2]. Furthermore, in TFLN structures, HF may impact the buried oxide layer as well and, hence, endanger the bonding. In addition, x-cut LN which is often used due to the above-mentioned advantages in domain fabrication, cannot be readily analyzed via selective etching. As HF etching affects the FE z-axis of the LN crystal only, it is necessary to expose the respective crystal surface by an additional step, e.g., via focused ion beam milling (see Figure 1b), a process step that dramatically limits fast fabrication. Finally, these adaptions lead to a low information content of the analysis: Rather than a large-scale domain pattern, selective TFLN etching only reveals the one-dimensional duty cycle and poling period information along the exposed z-face of the thin film. Hence, only small areas (<40 µm) of poled domain structures can be investigated in this way in x-cut TFLNs [1,5].

Hence, for the investigation of domain structures in TFLN, the non-destructive scanning probe or optical methods are preferred, which in addition, do not require any special sample preparation. As shown from previous investigations, SHG microscopy and piezoresponse force microscopy (PFM) can be successfully applied to inspect and analyze TFLN [1–3], analogously to their use for poled bulk LN [2,3,14,21,23]. Both methods directly detect a change in a material property which is connected to the FE domain orientation, namely the piezoelectric and nonlinear optical tensor, respectively. Hence, no special sample preparation for imaging is required. While to date, the specifics of the SHG contrast mechanism in TFLN have been thoroughly analyzed both by simulation and experiments, PFM has only been used as a mere imaging technique on TFLN, but no attempts have been

undertaken to further analyze the obtained signals, for instance, by explaining apparent differences between bulk LN and TFLN imaging.

Raman spectroscopy (RS) constitutes another (nonlinear) optical method which is well suited for imaging FE domain structures. While commonly applied to visualize bulk LN domain structures [24–28], no other works, to the best of our knowledge, so far have ever reported on the RS analysis and imaging of domain structures in TFLN. RS not only allows giving clear pictures of the domain distribution, but is also sensitive to other effects, such as defects, electric fields or stress distributions [29–31] which are often related to DWs. The potential to gain further insights into the TFLN system makes RS a promising and complementary tool for TFLN imaging and spectroscopy.

To date, most studies on PP-TFLN always rely on interpreting the results obtained from one single technique used for visualizing and analyzing the fabricated domain structures; comparing the different responses by complementary methods from one and the same sample spot is not reported yet. Here, a systematic comparison would allow not only to validate the potential and strength of every such method involved, but, furthermore, also to reveal the true domain shape and characteristics through cross-correlating the different contrast mechanisms of each technique. Such a comparison is more than relevant and needed, as contrast mechanisms and imaging results might always be influenced by the specific conditions of both the thin-film system and setup, as shown, for example, by SHG microscopy [23].

Therefore, we will combine here PFM, SHG, and RS for the analysis of x-cut TFLN down to the nanometer resolution. We will start by briefly reviewing the capabilities of every method and identify their strengths and weaknesses in order to determine the key performance factors in TFLN. Furthermore, we will identify the challenges for each method that may be addressed in future work.

## 2. Materials and Imaging Methods

### 2.1. Sample

The sample for the study here was fabricated from commercial ion-sliced TFLN (congruent 5% MgO-doped LN; NanoLN, Jinan Jingzheng Electronics Co., Ltd., Jinan, Shandong, China). The sample consists of a 300-nm thin single-crystalline lamella of a single FE domain oriented along the x-axes (x-cut TFLN), chip-bonded to a $SiO_2$/Si substrate (1800 nm $SiO_2$ on 500 μm Si {100}) to yield a lithium niobate on the insulator (LNOI) platform. In x-cut LN, the spontaneous polarization points parallel to the sample surface (in-plane polarization; z-axis). Therefore, electric field poling is readily achieved through electrodes manufactured solely onto the sample's top surface, as depicted in Figure 1a. The poling electrodes were structured with standard photolithography and lift-off, and consist of a 10-nm Cr adhesion layer and a 100-nm thin gold layer. Poling electrodes on the sub-μm sample in Section 3.2 were patterned via electron beam lithography. For poling, electrodes were contacted via electrical probes and a single, high voltage poling pulse surpassing the coercive field was applied. Details on the fabrication process and protocol are reported elsewhere [13,21].

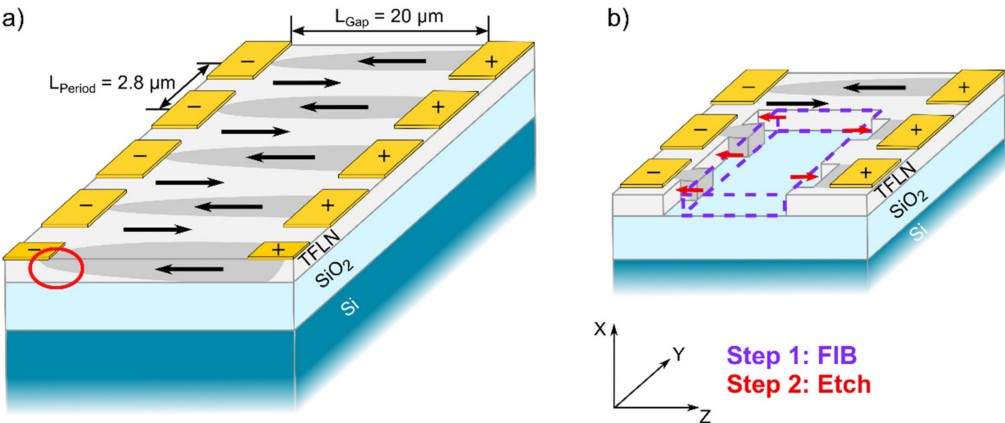

**Figure 1.** (**a**) Sketch of the periodically-poled x-cut thin film lithium niobate (TFLN) sample, consisting of a 300 nm lithium niobate (LN) thin film on a silicon substrate with silicon oxide (1800 nm). TFLN polarization is inverted by surface Au electrodes (period: 2.8 µm, gap distance: 20 µm). The inverted domains (dark grey) might not reach across the full depth of the LN thin film, as indicated in the foremost domain (red circle), since inversion along the *z*-axis proceeds faster than along x. (**b**) Selective etching scheme on x-cut TFLN. After exposure of the LN z-face by focused ion beam (FIB) (purple dashed box), subsequent etching with a higher etch rate on the negative polar face reveals the domain pattern, which is then ready to be visualized by standard imaging techniques.

## 2.2. Imaging Techniques

### 2.2.1. Piezoresponse Force Microscopy—PFM

Piezoresponse force microscopy (PFM) is a scanning probe technique based on atomic force microscopy (AFM). PFM is considered one of the standard techniques for imaging and analyzing FE materials and their domain structures [32–34]. PFM makes use of the fact that the orientation of FE domains directly determines the orientation of the piezoelectric tensor. Hence, by mapping the piezoelectric response, i.e., its phase and amplitude, FE domains can be visualized. Moreover, when quantifying the whole piezoelectric tensor in crystalline ferroelectrics, PFM may deliver the crystallographic sample orientation at the 1-nm length scale [35]. A typical PFM setup is sketched in Figure 2a. Here, an AC-voltage is applied to the conductive AFM cantilever tip, which is in contact with the grounded sample. The high electric field at the tip apex leads to a piezoelectric response of the sample, e.g., micromechanical expansion or contraction, of the piezoelectric material. This leads to an accompanied motion of the AFM tip, e.g., normal (deflection), lateral (torsion) [36] or buckling motion, which is then monitored by the AFM setup. The resolution of a PFM setup directly scales with the AFM tip diameter [37], but may be influenced by other parameters as well, such as the applied peak voltage or sample properties, e.g., depolarization charges [38]. In principle, a lateral resolution down to the 1-nm range is possible. More in-depth reviews on PFM, its capabilities, and different operation modes can be found in various reviews [32–34].

The PFM images in this study were recorded using a commercial Cypher AFM (Asylum Research, Oxford Instruments) and its built-in Vector PFM mode that allows analyzing lateral, as well as normal piezoelectric motions. For imaging, standard Pt-coated silicon tips were used (tip radius < 50 nm). For high-resolution imaging down to the 10-nm length scale, we used full-metal tips (Rocky Mountain tips; Pt-Ir; tip radius < 20 nm).

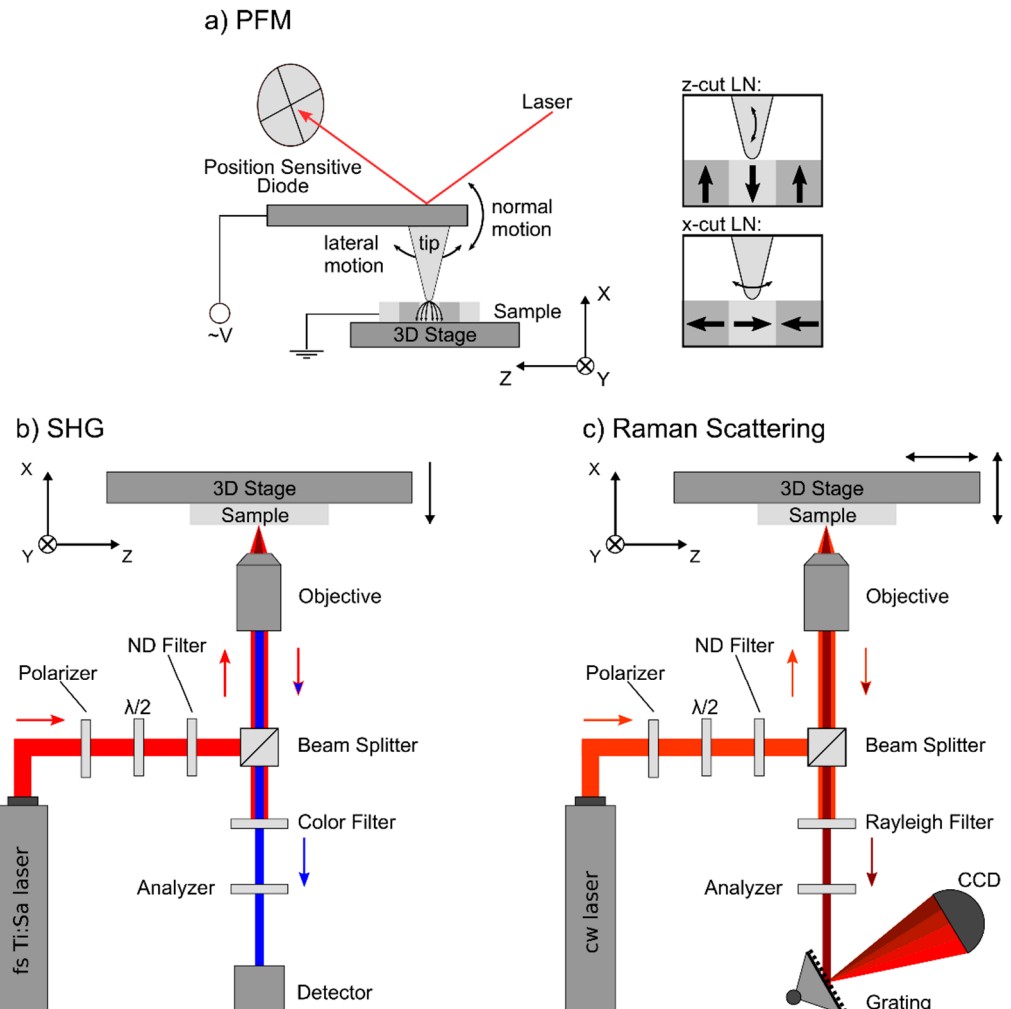

**Figure 2.** Schematic drawings of the different tools and setups as applied for imaging x-cut TFLN in this work. (**a**) In piezoresponse force microscopy (PFM), the normal or lateral motion of the cantilever tip is induced by the piezoelectric response to an electric field applied between the conductive tip and (insulating) sample. Insets to the right: Polarization directions and respective main tip motions for t-cut and x-cut LN samples. (**b**) In second-harmonic generation (SHG), the fundamental laser light is focused onto the sample through an objective, while the backscattered SH signal is collected via the same objective. Light with the fundamental wavelength is blocked via color filters, allowing only the second-harmonic signal to be detected. (**c**) The cw laser in the Raman backscattering setup is polarized, focused, and recollected analogously to (**b**). The Rayleigh filter blocks off most incident light (Rayleigh scattering), while the scattered "Raman" light is spectrally analyzed using a grating and appropriate charge-coupled device (CCD) camera.

### 2.2.2. Second-Harmonic Generation Microscopy—SHG

Second-harmonic generation (SHG) microscopy is an optical microscopy technique that relies on the second-order nonlinear optical effect of second-harmonic generation as its contrast mechanism. In the SHG process, two photons at the pump wavelength are annihilated to create a single photon at half the wavelength (twice the energy of a single photon). Analogously to PFM, SHG and hence the SH tensor are dependent on the crystal symmetry, i.e., FE domains and their orientation. Therefore, it can be ideally used to visualize domain structures. Due to its nature as a nonlinear, optical process, it requires high pump intensities usually provided from ultra-fast, fs-pulsed laser light sources.

For this work, we used two different SHG setups, one self-built SHG microscope, as well as a commercial microscope (SP5 MP by Leica Microsystems, Wetzlar, Hessen, Germany). Figure 2b schematically depicts their optical layout. Both SHG systems use a tunable fs-Ti: Sapphire laser as the pump source (self-built: Griffin by KMLabs, Boulder,

Colorado, USA, tuning range: 760–840 nm; Leica: MaiTai by Spectra-Physics, Stahnsdorf, Brandenburg, Germany, tuning range 720–900 nm). The pump light is directed to the sample via a dichroic mirror and a high numerical aperture (NA) air objective (self-built: NA = 0.9, Leica: NA = 0.8). The locally generated SH light is collected in back-reflection through the same objective. To block off any pump light from the detector (self-built: Silicon single photon avalanche diode; Leica: Photo-multiplier tubes) appropriate short-pass filters are inserted into the detection path. Since SH light is only generated point-wise at the focal location, a scanning procedure is needed in order to generate SHG images; hereto galvo mirror scanners steering the pump beam are installed in the Leica microscope, whereas the sample is mounted on a 3D-scanning piezo stage with a fixed focus in the home-built setup.

### 2.2.3. Raman Micro-Spectroscopy—RS

The Raman effect describes the inelastic scattering of photons with low energy excitations. In crystalline media, Raman scattering (RS) is primarily linked to the excitation of phonons. Scattering properties such as the phonon frequency or the scattering cross section are very sensitive to changes in the crystal structure, i.e., the geometry of the crystal lattice and its atomic composition. As vibrational motions and involved atoms significantly vary between different phonons, the specific influence of structural changes on certain phonon properties can give valuable insights into the nanoscopic impacts of microscopic effects, such as the position of DWs, which represent a specific change in the crystal structure. Hence, by spatially mapping changes in the phonon spectrum, DWs and other influences can be readily distinguished.

RS experiments are carried out here using a Horiba LabRAM HR Evolution commercial Raman spectroscope (HORIBA Jobin Yvon GmbH, Bensheim, Hessen, Germany), ($\lambda$ = 633 nm, P = 17 mW cw, 1800 L/mm grating). A scheme of the used backscattering setup is shown in Figure 2c. The high NA = 0.9 of the microscope objective provides a good signal strength and a small focal spot (lateral Rayleigh criterion: 429 nm; axial Rayleigh criterion: 1.56 μm) to detect signals mainly from TFLN. Spectra were recorded in x(zz)-x ($A_1$(TO) modes) and x(zy)-x (E(TO) modes) Porto geometries [20]. To ensure a constant focus for the whole measurement, an automated focusing routine is applied to every measuring spot, consisting of a depth sweep with an automated fitting procedure, in order to allocate that focus depth at which the largest signal intensity for the $A_1$(TO$_1$) peak (~250 cm$^{-1}$) or the E(TO$_1$) mode (~155 cm$^{-1}$), respectively, is recorded. Images are generated by analyzing phonon peak properties (peak intensity, frequency shifts) of hyperspectral Raman maps. In this work, a lateral point-to-point separation of 100 nm in y-direction and 500 nm in z-direction was chosen with a higher sampling rate along the *y*-axis, i.e., perpendicular to the created domain walls.

## 3. Results and Discussion

To compare the potential and restrictions of each imaging technique listed in Section 2, the PP-TFLN sample with a domain period of $\Lambda$ = 2.8 μm is analyzed with PFM, SHG, and RS. A selection of the results is displayed in Figure 3. Here, for each method, we selected a signal property that provides a DW contrast. All images show a similar representation of the domain structure, demonstrating that every method in principle is able to visualize the same structure, despite the fact that each method probes different material properties, i.e., the piezoelectric tensor for PFM, the nonlinear tensor in SHG, and the phonon spectrum for RS. However, the colormaps differ in properties such as the signal-to-noise ratio, number of pixels, and most importantly the perceived DW width. Partly, this is due to the different spatial resolution, but also since different material properties change at different length scales close to DWs [19,27]. The respective imaging details will be discussed in the following sections.

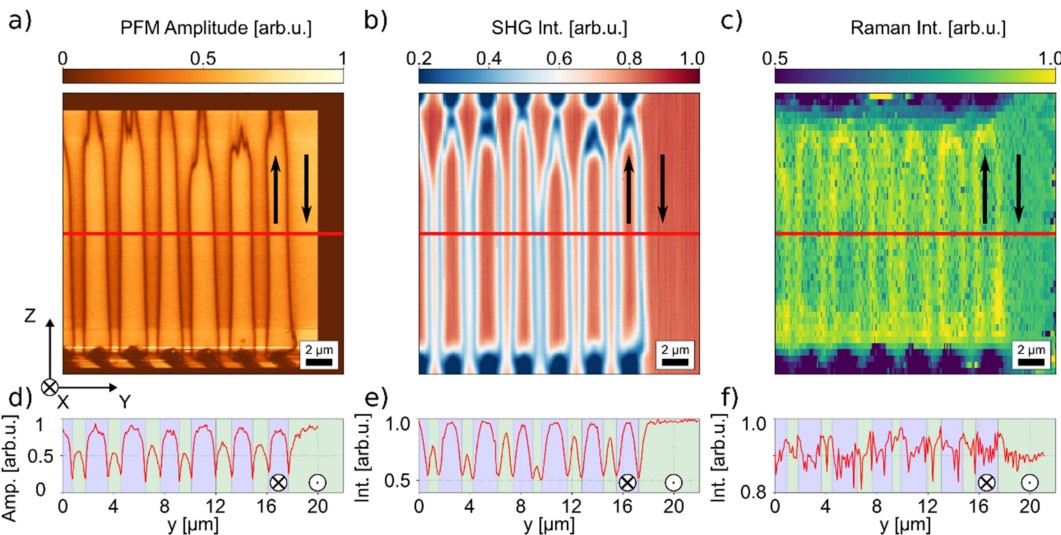

**Figure 3.** Periodically-poled thin film lithium niobate (PP-TFLN) in-plane ferroelectric domains imaged consecutively by (**a**) piezoresponse force microscopy (PFM), (**b**) second-harmonic generation (SHG), and (**c**) Raman spectroscopy (RS) over exactly the same sample area. Note the pronounced domain wall (DW) contrast by all techniques. The colormaps in (**d**–**f**) show cross-sections taken along the indicated red lines of the respective method (**a**–**c**). As a guide to the eye, green (pristine) and blue (inverted) domain areas mark the differently oriented domains. All measurements show the same domain pattern despite their different contrast mechanisms.

### 3.1. Piezoresponse Force Microscopy

PFM is a widespread method for the analysis and imaging of FE thin films or bulk crystals [32–34]. The contrast in PFM relies on directly probing the piezoelectric tensor, which is connected to the FE domain orientation. In PFM, the tip is in contact with the sample, and an AC-voltage is applied between the conductive tip and (insulating) sample. Depending on the sample orientation, i.e., the piezoelectric tensor orientation, the sample will react with a piezoelectric deformation, e.g., contraction or expansion, to the electric field. In this simple picture, the domain contrast can be explained as follows: Lithium niobate is a uniaxial FE crystal, where only two domain orientations are allowed along its crystallographic z-direction. Here, the piezoelectric tensor is inverted in domains of antiparallel order parameter orientation. Hence, the piezoelectric response will be 180° phase-shifted between two such adjacent domains. In a simple Ising-type picture, the polarization on DWs in ferroelectrics shows a tanh shape with a width of a few unit cells [19]. The piezoelectric tensor is changed accordingly, with a magnitude of 0 in the center and maximum values within domains. Hence, a 180° phase shift is observed in the PFM phase, while the PFM amplitude will show a minimum at the DW center. In principle, the limiting factor in the resolution in PFM is the tip radius. Hence, resolutions down to single digit nm have been reported [19]. In real experiments, the observed resolution is often limited by additional factors, such as the applied electric field, sample geometry, the dielectric properties, screening charges in the bulk or at the sample surface [38]. In depth reviews on PFM can be found elsewhere [32–34,38].

Figure 4 shows a typical PFM result on our x-cut TFLN sample, with a-c displaying the topography, the PFM phase and PFM amplitude, recorded simultaneously. As expected, the topography image exhibits only weak features with heights <0.5 nm that are not correlated to the domain structure. In contrast, the electrodes are visible at the lower sample edge with heights of >50 nm. The phase signal provides a net domain polarity contrast of 180° between antiparallel domains, as expected. Conversely, the amplitude signal shows a dominant DW contrast with a signal width (FWHM) of <200 nm. To create images such as these, typical scan rates around ~1 Hz per line are recommended, leading to a total acquisition time per frame of minutes when using scan increments of 100 nm.

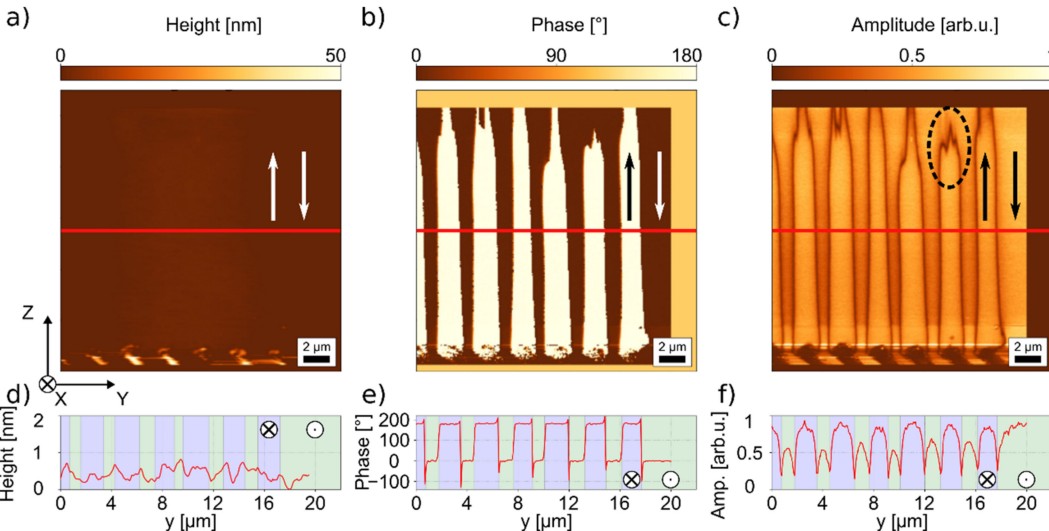

**Figure 4.** PFM colormaps of (**a**) topography, (**b**) phase, and (**c**) amplitude signal for an x-cut PP-TFLN sample, and (**d**–**f**) respective line scan graphs taken along the red line in (**a**–**c**). While the height signal shows no topographical features in poled regions, the phase signal provides domain inversion, while the amplitude signal allows imaging with a superb DW contrast. The circled area in the amplitude map (**c**) might be indicative for an incomplete in-depth domain inversion close to the (−) electrode (see text).

The high lateral resolution makes PFM the method of choice when it comes to imaging and detecting sub-micrometer polar features, as shown in Figure 5 for a TFLN sample with a periodicity of Λ = 0.8 µm. Such samples are not accessible with diffraction-limited (optical) methods, as these periodicities lie in the range of the Abbe limit. Similar to Figure 4, the results on the submicron structure in Figure 5 show no domain related contrast in the topography. While the PFM phase shows a domain polarity contrast, the PFM amplitude allows distinguishing between individual domains with DW-DW distances of less than 200 nm and individually resolved DWs (see Figure 5b,d).

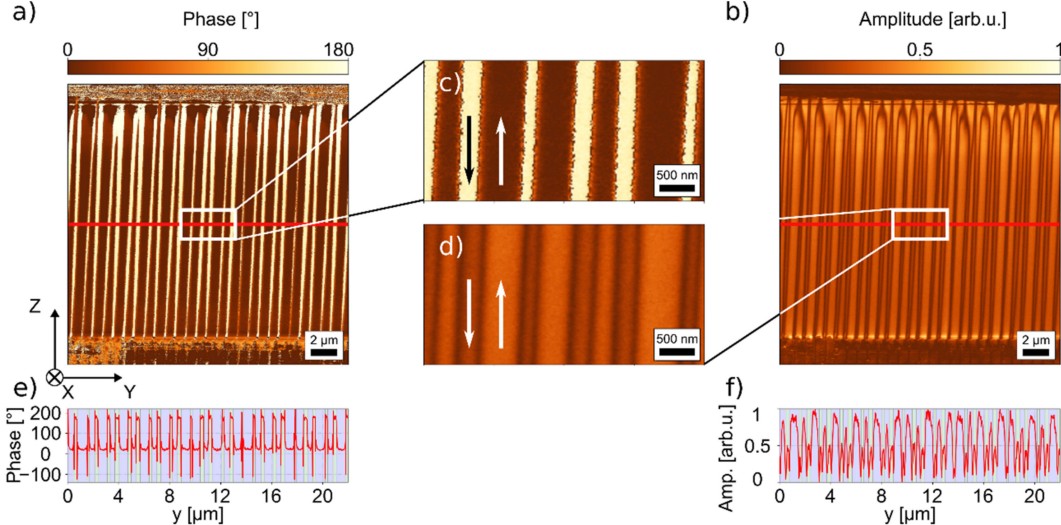

**Figure 5.** PFM colormaps of an x-cut PP-TFLN sample with sub-micrometer periodicity. (**a**) Phase map providing the domain polarity contrast. (**b**) Amplitude map with the domain wall contrast. (**c**) and (**d**) Zoomed-in map segments of the phase (**c**) and amplitude (**d**) maps that demonstrate high-resolution imaging of sub-µm domains. (**e**) and (**f**) Line scan graphs of (**e**) phase and (**f**) amplitude maps along the red lines indicated in the respective colormaps.

In principle, PFM is also sensitive to vertically stacked domains in the depth of the sample. If domains of different orientations are stacked upon each other, their piezoelectric response will partly cancel. Hence, a decrease in PFM amplitude may be expected. However, as the electric field rapidly decays with the increasing distance away from the tip, i.e., typically with $1/r$, where r is the tip radius, PFM provides only a limited and nonlinear depth resolution. Experiments performed on vertically stacked domains in z-cut bulk lithium niobate indicated that domains up to a depth of 1.7 μm could be detected with a normal (deflection) mode (90% criterion) [39]. This suggests that in principle PFM allows imaging the domain constitutions into the full depth of these TFLN samples, as film thicknesses typically measure 300–700 nm. Indeed, a comparison of the SHG and PFM images in Figure 3 suggests that some of these domains might not be fully inverted in the full TFLN sample depth, as deduced from the reduced PFM amplitude of the encircled domain in Figure 4c. As seen close to the top sample edge where the negative electrode for poling is located, the color contrast is slightly shaded as compared to the fully poled domains. This information is corroborated between both the PFM and SHG information. Nevertheless, further experiments and theoretical analysis are necessary to calibrate the depth resolution in PFM for this type of sample, since the amplitude does not decrease linearly with the domain depth. Furthermore, in clear contrast to previous depth-resolved PFM experiments on z-cut bulk LN, x-cut TFLN features a different crystal orientation. Hence, smaller piezoelectric tensor elements and/or different cantilever motions are addressed. Moreover, the used substrate material stack will certainly influence the electric field distribution, as well.

In conclusion, PFM unambiguously allows imaging the FE domain structures due to the well-defined signal origin and behavior. The advantage of PFM is its capability of high-resolution imaging since it is not limited by diffraction. This makes PFM the best-suited method for imaging micrometer- and sub-micrometer-sized structures. However, due to its limited scan speed, visualization by PFM is limited to smaller areas. Hence, for analyzing mm-sized PP-TFLN structures as requested for narrow-band and highly efficient optoelectronic applications, PFM needs to be complemented by additional techniques, e.g., SHG imaging. In z-cut TFLN, PFM has another particular advantage, as it also allows writing and subsequently analyzing FE domain structures. Recently, this has seen widespread use for studies of conductive domain walls [40,41] in TFLN.

### 3.2. Second-Harmonic Generation Microscopy

Second-harmonic generation microscopy sees widespread use for imaging and analyzing FE domain structures. It is frequently applied to bulk samples, since it allows fast, large-scale, and non-destructive 3D imaging with a diffraction-limited optical resolution [25,42–45]. The high speed of this technique facilitates in situ imaging of FE domains and domain walls in bulk samples under external stimuli, such as electric fields or temperature [46–49]. In SHG microscopy, the contrast mechanism relies on probing the second-order susceptibility tensor $\chi^{(2)}$, whose properties are directly connected to the FE domain orientation. Lithium niobate is a uniaxial ferroelectric. Here, only two domain orientations are allowed. The tensor $\chi^{(2)}$ is inverted, i.e., switched from $+\chi^{(2)}$ to $-\chi^{(2)}$ between antiparallel domains. The SHG process is a coherent, optical process. Whenever the sign of the nonlinear tensor $\chi^{(2)}$ is reversed, the generated SH light will see an additional $180°$ phase shift compared to the SH light generated in a region with a non-inverted nonlinear tensor. According to the experimental and theoretical analysis, FE DWs are only a few unit cells wide [19], i.e., significantly smaller than the diffraction limited optical focus spot (<500 nm diameter). Neglecting any other effects accompanying a DW (which may further influence the SHG response, e.g., strain fields, defects or electric fields) this allows explaining the appearance of DW in a SHG image in the simple picture. Assuming a focus is placed symmetrically on top of a domain wall, the light generated within one side of the DW will destructively interfere with the SHG signal on the other side of the DW, due to the inverted tensor. Hence, a DW will appear as a dark line within an SHG image. This

simple picture holds true, if a tensor element can be directly addressed with the pump light polarization, and SHG imaging is performed near the surface (correct beam focusing, normal dispersion, i.e., $n_{SHG} > n_{fund}$). It should be noted that the contrast mechanism may be modified for large numerical apertures [50] or that other contrast mechanisms, e.g., the so called Čerenkov contrast or modified SH tensors at DWs, can play a significant role in bulk systems. However, as theoretical and experimental works suggest [23], these effects can be neglected for x-cut thin films, since the largest tensor element $d_{33}$ can be directly addressed, and imaging is always near the surface (i.e., film thicknesses and focus depth are similar). Therefore, SHG microscopy is well suited for non-destructive, large-scale imaging of periodically-poled structures.

Due to its nature as a coherent optical process, SHG microscopy is further sensitive to the inversion depth well below the optical depth of focus. If domains of opposing directions are stacked vertically, the signal generated from the opposing domains will interfere destructively resulting in a reduced signal as compared to the fully inverted or non-inverted single domain. As SHG scales quadratically with the interaction, this allows sensitively detecting the stacked domains with an accuracy of tens of nanometers [23]. This is possible for TFLN, since the coherent interaction length of SHG in the forward direction ("co-propagating" phase matching) measures about 1.28 µm for 800 nm fundamental light, which is larger than the typical film thickness used in integrated optics (<1 µm). The light generated in the forward direction can be measured even in the backward detection due to the high reflectivity of the substrate. In SHG, the light is also directly generated in the backward direction. However, due to the short coherent interaction length (~45 nm) and the quadratic scaling with the interaction length, this counter-propagating light is two to three orders of magnitude weaker in intensity. It should be noted that using a nonlinear substrate, e.g., LN rather than a silicon wafer, may significantly influence the detected SH light, since the SH light is not only generated in the film, but also in the substrate. Further insights into the SHG imaging process can be found in our previous paper [23].

To demonstrate the sensitivity for the inversion depth of SHG microscopy, we have performed numerical simulations of the imaging process for a typical x-cut film of 300 nm thickness on a 1.8 µm oxide layer, similar to our structure in Figure 1a. For simulation, we used the code that was previously developed [19,23]. Figure 6a depicts the simulation setup. Here, we assume an inverted domain of 2 µm width with a hexagonal cross-section, as it can be expected for x-cut films. For the simulation, the domain extends infinitely along the *z*-axis. Figure 6b shows the calculated focus field for a NA of 0.8 and a wavelength of 800 nm, which is used for all simulations reported in this paper. Figure 6b shows that reflections play a crucial role in the fundamental field. As shown previously, similar conclusions can be drawn for the reflected SHG signal, which is the main SHG signal that is detected. In our simulations, the focus is moved in increments of 50 nm and the reflected SHG intensity collected by the objective lens is calculated. This simulation was performed for different depths of inverted domains in 50 nm increments, and the result is plotted in Figure 6c. These simulations show that only for a fully inverted domain (h = 300 nm), the SHG signal level within the domain recovers and reaches up to the bulk value (h = 0 nm). For this case, only vertical DWs lead to a signal decrease, i.e., a domain wall contrast is observed. The simulation demonstrates that due to the coherent nature of the process, domain inversion depths of much less than the optical resolution can be detected by SHG.

To show that the contrast and imaging mechanism is independent of the setup, e.g., NA or wavelength, we have imaged the same area with our two setups and at two different wavelengths. These results are displayed in Figure 7. Overall, a similar domain image is obtained independent of the setup. In detail, the resolution decreases from (a) to (c), which is a result of the reduced NA (self-built: NA = 0.9, Leica: NA = 0.8) and the increased wavelength (c). To realize an image as shown in Figure 7a, the self-built system requires an integration time typically on the order of 10 to 50 ms. Hence, assuming a step width of 200 nm along each direction, typical measurement times in the order of 10–30 min are required with the self-built setup. Due to the high dynamical range of the photo-multiplier

tube and the beam scanning rather than sample scanning, the commercial system only requires integration times of much less than 1 ms per pixel. Complete frames, as shown in Figure 7b,c, can be visualized within a second or less, allowing for fast imaging. Via moving the sample on a piezo stage and stitching these frames, large images (>1 cm$^2$) can be created in a reasonable time.

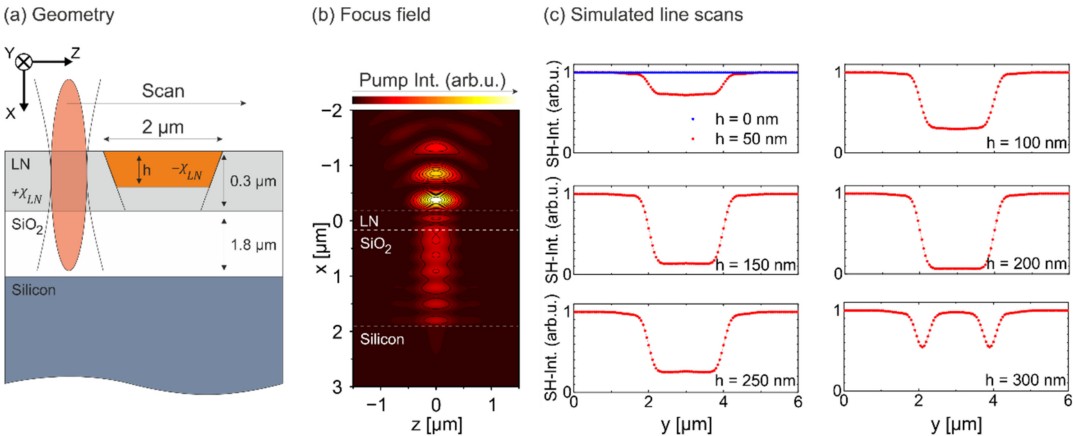

**Figure 6.** SHG simulations for x-cut TFLN with NA = 0.8 and $\lambda_0$ = 800 nm. (**a**) Sample geometry and simulation layout. (**b**) Focus field simulation for all sample layers. (**c**) Simulated SHG intensities for different poling depths of inverted domains in increments of 50 nm domain depth. Only a completely inverted domain allows imaging with a clean DW contrast.

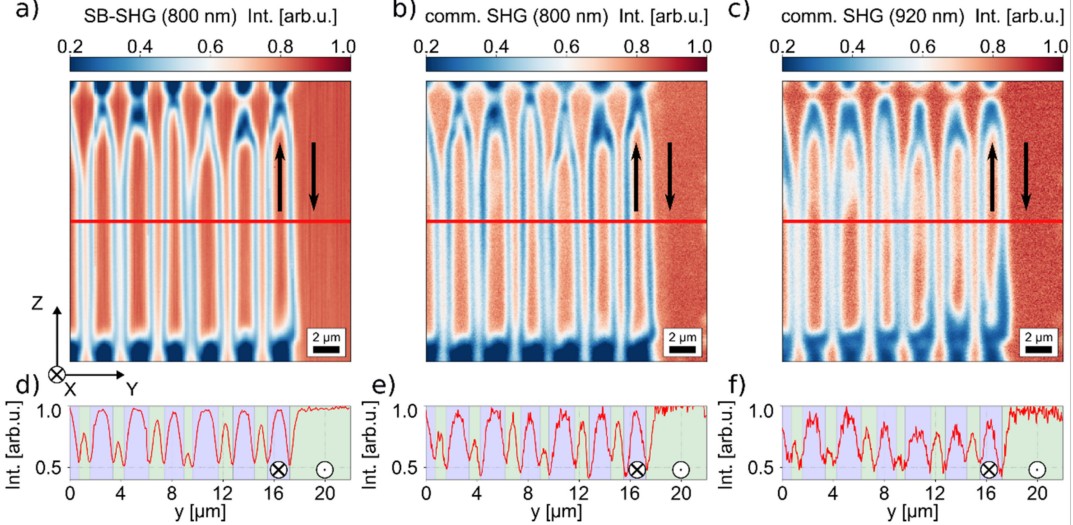

**Figure 7.** SHG micrographs obtained with (**a**) the self-built setup (NA = 0.9)typical measurement times in the order o, and (**b**) the commercial Leica SP5 MP (NA = 0.8), both recorded at an incident wavelength of 800 nm. (**c**) SHG image with 920 nm incident light measured with the Leica SP5 MP. The DW contrasts of all images are comparable, indicating no dependencies on the wavelength or setup over the analyzed wavelength range. (**d–f**) Line scan graphs of the maps (**a–c**) along the red lines indicated in the respective colormaps.

To demonstrate the capability of SHG for large-scale imaging of complete chips, we have visualized the domain structures along a full 5 mm electrode with the Leica microscope. The result is displayed in Figure 8a. In Subfigure b–d, detailed zoomed-in images taken from the large scan are shown. The zoomed-in frames exhibit similar features as the images in Figure 7. This means that even at this large scale, no loss of information is observed and the poling quality, e.g., poling period, homogeneity, duty cycle or poling depth, can be analyzed for a full wafer. The large image in (a) is cut from a stitched image of 20 frames, each with a size of approximately 260 × 260 µm$^2$. To account for the non-perfectly planar mounting of the TFLN sample (approximately 50 µm height

difference along 5 mm), a depth scan over 50 μm with a step size of 5 μm was conducted for each frame. Due to the slightly different focus in each frame, slightly different overall intensities are observed, leading to the large-scale intensity oscillations observed along the chip. As the previous theoretical analysis indicates, the focus position does not influence the domain wall contrast [23]. Taking the images at all different planes into account, a total of 200 frames were scanned for this image. Assuming an imaging time of ~1 s per frame and accounting for (automatic) sample repositioning, a total imaging time in the range of only a few minutes is required for the example shown here. The scan speed may be further increased though at the cost of signal-to-noise ratio. Whenever a low magnification objective lens is used with a larger field of view, even higher imaging speeds are possible at the cost of lateral resolution. This example demonstrates that complete chips (~cm$^2$) can be imaged within an hour or even less.

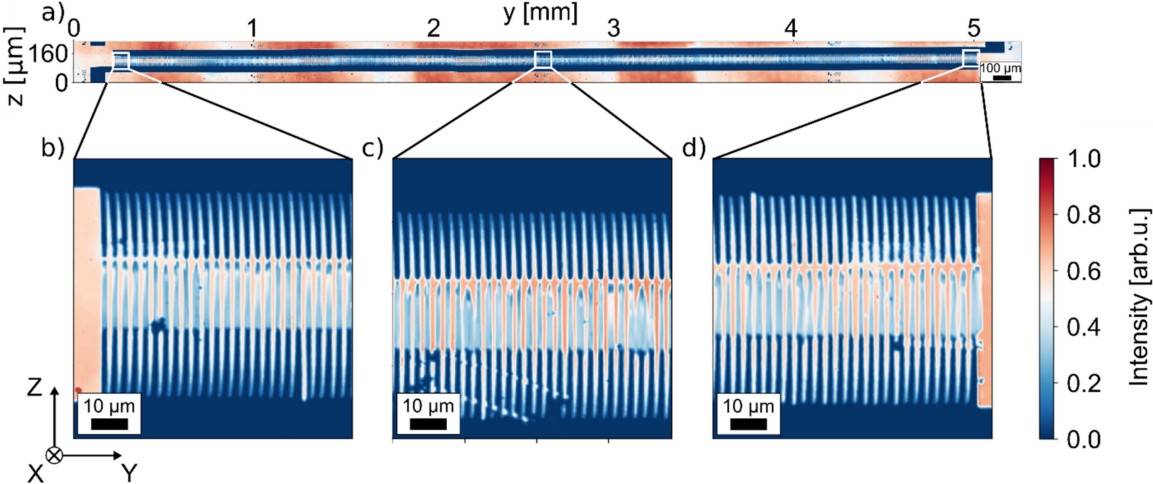

**Figure 8.** Large-scale SHG imaging of the complete, 5-mm-long PP-TFLN sample. (**a**) Overview micrograph obtained by stitching 20 individual measurements together. (**b**–**d**) Detailed high-resolution scans at different locations along the waveguide structure.

As demonstrated, the imaging process and contrast mechanism of SHG microscopy in TFLN can be well understood based on simple assumptions. Nevertheless, there are several open questions that need to be addressed in future works. A key aspect of our model is that FE DWs are assumed to constitute a Heaviside function flipping instantaneously from $+\chi^{(2)}$ to $-\chi^{(2)}$. Based on this model, the contrast can be solely explained by an interference process of SH light generated in opposing domains. As we have shown, this model may well explain our observations made in x-cut TFLN, where also the largest tensor element is addressed. However, various works on FE DWs demonstrated that DWs exhibit an internal structure and are accompanied by other effects, such as strain or electric fields [19], which have been demonstrated to span over several μm around the DW and are likely to have an influence on the local second-harmonic response in the vicinity of the DW [51]. In this context, techniques such as SHG polarimetry allow proving the existence of deviations from the ideal Ising wall in ferroelectrics [42]. Therefore, SHG polarimetry is indicated for TFLN in future work. This might help better understand the fundamental physics of domains in TFLN, e.g., the nature of the buried h2h and t2t DWs close to the electrodes. An additional substructure of $\chi^{(2)}$ can be readily included in the numerical model to provide further insight into DW properties in TFLN. In the context of electric or strain fields, correlative imaging with multiple methods, e.g., RS or PFM, might give a further understanding of the influence of these fields, as well as on the SHG properties, which will more accurately allow for the interpretation of measurements.

In conclusion, SHG provides fast imaging and analysis of domain structures in TFLN devices. In principle, high resolution images of complete TFLN chips (~cm$^2$) containing

hundreds of poled areas can be imaged within less than an hour when using modern microscopes, while detailed images of smaller areas ($<500 \times 500$ μm$^2$) can be taken in a matter of seconds. For z-cut TFLN or bulk samples, this imaging speed can only be matched by selective etching and subsequent microscopy imaging, yet at the cost of sample destruction. Even more, the key advantage of SHG imaging on TFLN is that it can unambiguously distinguish between fully or only partly penetrated domains with sub-diffraction-limited depth resolution, providing valuable information in the performance analysis. Due to the non-destructive nature and high speed, SHG microscopy can provide valuable information to design and fabrication, e.g., by selecting only suitable areas within the electrodes for device fabrication.

*3.3. Hyperspectral Raman Imaging*

Raman scattering microscopy is a common technique for the investigation of crystal structures and has been widely used for imaging LN DWs [24–28]. The RS contrast of DWs in poled LN has been ascribed to two different mechanisms [27]:

(1)     LN DWs that are described as large two-dimensional defects by Stone and Dierolf, introduce a quasi-momentum to the RS process, which is directed perpendicularly to the plane of the DW. If measured on the crystal z-face, the detected signal originating from the DW does not stem from phonons propagating along the beam direction, but from obliquely propagating phonons. Therefore, the selection rules are lifted at DWs, and A$_1$(TO) phonons or mixed longitudinal optical-transversal optical (LO-TO) phonons can be detected, which are usually not accessible in that geometry [28].

(2)     In the vicinity of DWs, a large strain [52] and electric fields [53] have been observed. Based on these findings, Fontana et al. concluded that these fields change the RS efficiencies due to elastooptic or electrooptic coupling [54]. Capek et al. conclude that the observed phonon frequency shifts at DWs are also caused by local fields [26].

For Raman scattering on the incident x-face of bulk LN, no directional dispersion as in case 1 has been observed in the used scattering configurations [27,55]. Instead, the DW signature here presumably results only from phonon frequency shifts due to strain and electric fields, and intensity changes resulting from elastooptic and electrooptic coupling, as explained in case 2. In RS experiments on poled TFLN structures, we expect a similar behavior as for these investigations on non-polar surfaces of LN bulk samples, although possible large-area stress fields resulting from the bonding process of the LN thin film and the confinement along one dimension, can additionally affect the detected scattering signal and lead to different results.

Although a common analysis method for poled bulk LN, Raman micro-spectroscopy, to the best of our knowledge, has not yet been applied for TFLN imaging beyond our own earlier work [56]. This might be due to the increased challenges faced when inspecting thin films in general: Due to the low interaction volume of the thin film with the incident laser photons, the generated scattering signal is weak in absolute terms and also compared to the substrate signal. This fact calls for long acquisition times (in our measurement, we applied acquisition times per pixel of $2 \times 5$ s/px in x(zz)-x and $2 \times 150$ s/px in x(zy)-x), precise focusing and careful peak analysis.

The Raman spectra in a virgin domain and on a domain wall are compared in Figure 9. Only the four indicated phonon peaks are generated in the LN thin film, whereas the SiO$_2$/Si substrate causes a strong background with the characteristic triply degenerate Si peak at 521 cm$^{-1}$. For this reason, it is essential to keep the laser focus in the thin film and maximize the LN scattering signal, which is achieved by the autofocusing routine detailed in Section 2. This special treatment massively prolongs the measure time per pixel, which makes Raman investigations the slowest of the techniques shown in this work (the total acquisition time for the shown Raman colormaps including autofocus is ~15 days; the pure acquisition time without autofocus amounts to 27.6 h).

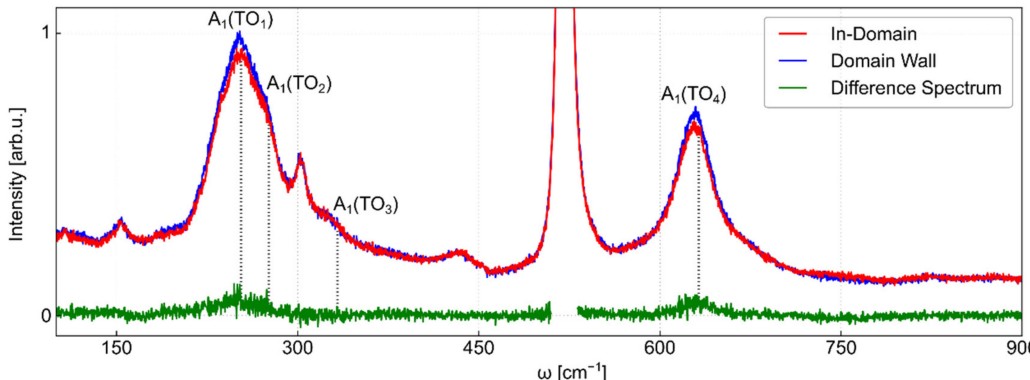

**Figure 9.** Comparison of Raman spectra taken inside a domain (red) and directly at the DW (blue) in x-cut TFLN on $SiO_2$/Si in x(zz)-x geometry. The expected LN phonon peaks are marked in the figure. As seen in the different spectrum (green), the detected LN phonon modes $A_1(TO_1)$, $A_1(TO_2)$, and $A_1(TO_4)$ experience spectral changes between the domain walls and poled domains. Due to the small scattering volume of the thin film, the phonon peaks of the Si substrate form a strong background.

For Raman imaging of the x-cut TFLN sample, we analyzed the intensities, frequencies, and widths (FWHM) of $A_1(TO)$ and $E(TO)$ peaks. A complete collection of $A_1(TO)$ spectral features in TFLN is provided in the Supplementary Materials. The TFLN phonon features partly deviate from the bulk behavior, as a comparison of the $A_1(TO_4)$ features shows. Whereas only the peak frequency analysis yields a notable DW contrast in bulk LN [20], TFLN DWs are accessible via the frequency, scattering intensity, and FWHM of $A_1(TO_4)$. This different behavior might be attributed to the influence of the substrate bonding on the acoustooptic coupling and a stronger inhomogeneity of the local stress distribution due to the low film thickness.

Figure 10 shows a selection of hyperspectral colormaps generated from $A_1(TO)$ phonon features with different contrasts. Imaging with the domain wall (Figure 10a) as well as the domain polarity contrast (Figure 10b) can be achieved by addressing different phonon properties from the same measurement. The different phonon responses can be explained by the atom motions associated with the respective phonons [57,58]. Whereas the dominant motion in $A_1(TO_4)$ is a vibration of the oxygen hexagon, the $A_1(TO_2)$ phonon is dominated by a vibration of lithium atoms along the *z*-axis. As the polarity change in inverted domains is caused by a dislocation of niobium atoms within their respective oxygen cages as well as hopping of lithium atoms into adjacent oxygen cages, the change of polarization directly affects the vibrational motion of the $A_1(TO_2)$ phonon. On the other hand, the atom movement in $A_1(TO_4)$ is not affected by the specific polarization direction, but only in the transitional region, i.e., in the vicinity of the DW. Therefore, $A_1(TO_2)$ phonon properties provide a domain polarity contrast, whereas $A_1(TO_4)$ phonon properties enable a DW contrast.

Paying closer attention to the DW contrast mapping in Figure 10a (e.g., in the area marked by the dashed circle), the image shows similar features such as the SHG map in Figure 7c, which is subject to poling depth investigations. However, different from SHG, we assume that this signal does not allow accessibility to the poling depth, but results from the presence of buried domain walls. The Raman signal is generated in full depth of the illumination spot and hence is sensitive to structural changes in full depth of the LN thin film. Whenever a poled domain does not extend to the full depth of the thin film, a DW in the y-z-plane is generated. This DW is accessible via Raman imaging mechanisms that are sensitive to DWs, such as $A_1(TO_4)$ phonon properties. Therefore, the Raman scattering investigation may be facilitated to monitor the completeness of domain in-depth poling, however, does not yield quantitative information on the poling depth.

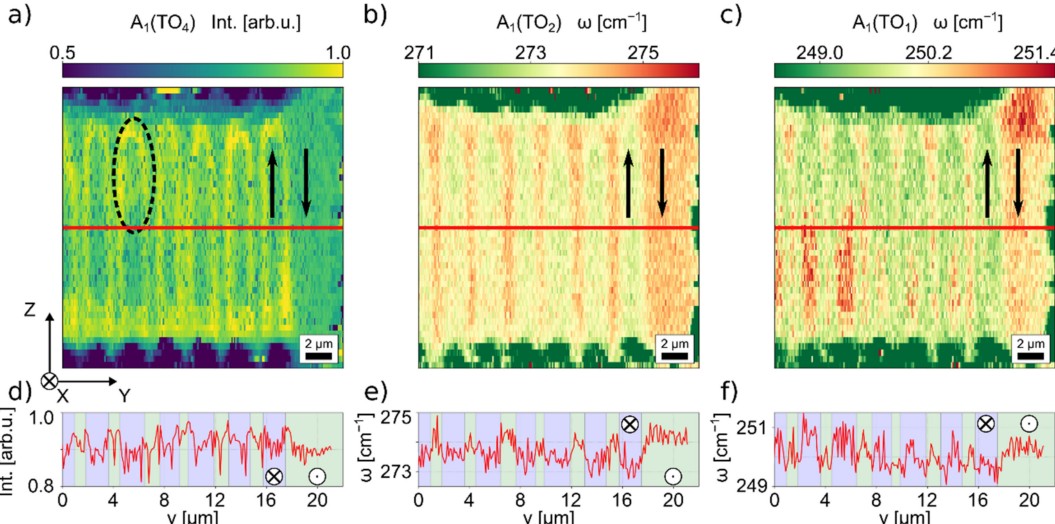

**Figure 10.** Hyperspectral Raman colormaps of an x-cut PP-TFLN sample (**a–c**), and (**d–f**) the respective cross-sections taken along the indicated red lines in (**a–c**). The step size of the measure points is 0.5 μm along the TFLN z-axis and 0.1 μm along the y-axis. (**a**) The $A_1(TO_4)$ intensity provides a net DW contrast with a suspected sensitivity to check for poling completeness, for instance, as in the circled area. (**b**) The $A_1(TO_2)$ frequency reveals a domain polarity contrast. (**c**) The frequency of $A_1(TO_1)$ also shows the DW contrast with a suspected sensitivity to check for poling completeness, and is assumed to be sensitive to stress fields, as well.

A peculiar behavior can be seen for the $A_1(TO_1)$ peak frequency mapping in Figure 10c. Similar to the $A_1(TO_4)$ intensity, this feature shows a domain wall contrast with the assumed sensitivity for poling completeness, but in the unpoled region of the sample (upper right area of the colormap), inhomogeneities of the signal, i.e., a phonon frequency change of ~1.5 cm$^{-1}$, can be seen although there is no evident structuring. The cross correlation with PFM and SHG measurements shows that no structural features are to be expected in this area. A possible explanation for this observation is a sensitivity to mechanical stress and/or electric fields. Previous works on the pressure dependence of LN phonon features [30] suggest that the isotropic pressure dependence coefficient of $A_1(TO_1)$ is ~0 cm$^{-1}$/kbar, which would directly contradict this assumption. However, that work gives no information on the influence of uni- or biaxial stress fields. Stone et al. [29] ascertained the scattering dependence on apparent electric fields, but only listed data for $A_1(LO)$ and $E(TO)$ phonons. Assuming that the coefficient of $A_1(TO_4)$ is in a comparable order of magnitude such as $A_1(LO_4)$, the electric field is estimated to be in the order of 45 kV/mm. In both cases of strain and electric field dependence of the phonon frequency, future work is indicated to extend the fundamental knowledge on these dependencies and evaluate our assumptions.

An example of Raman imaging with $E(TO)$ modes is shown in Figure 11. Here, we have analyzed a different electrode of the same sample that provides larger domains, which allows a more clear visualization of the phonon frequency shift between oppositely-poled domains. The resulting larger domain sizes allow a clearer distinction of the phonon properties. The phonon frequency of the $E(TO_2)$ mode provides a clear domain polarity contrast. The frequency change between the oppositely-poled domains has been applied to the pressure and electric field correlation studies. With a pressure dependency coefficient of 0.05 cm$^{-1}$/kbar [30], the $E(TO_2)$ frequency change of $-0.27$ cm$^{-1}$ from pristine to poled domain would correspond to $-5.4$ kbar, i.e., a tensile force acting on the LN thin film due to the polarization change. With an electric field coefficient of $-0.01$ cm$^{-1}$/(kV/mm) [29], a change of electric field by 27 kV/mm between oppositely-poled domains can be deduced. While this number is larger than the coercive field in undoped, bulk LN [59], several works reported that for poling doped TFLN (which even has a lower coercive field) high electric fields need to be applied reaching values of up to 40–60 kV/mm in order to achieve successful poling. While some authors suspected this to be an effect of the poling

electrode design and to facilitate nucleation, our results indicate that these high fields maybe necessary to overcome internal limits, e.g., imposed fields and charges required to stabilize h2h and t2t DWs. Further research is necessary.

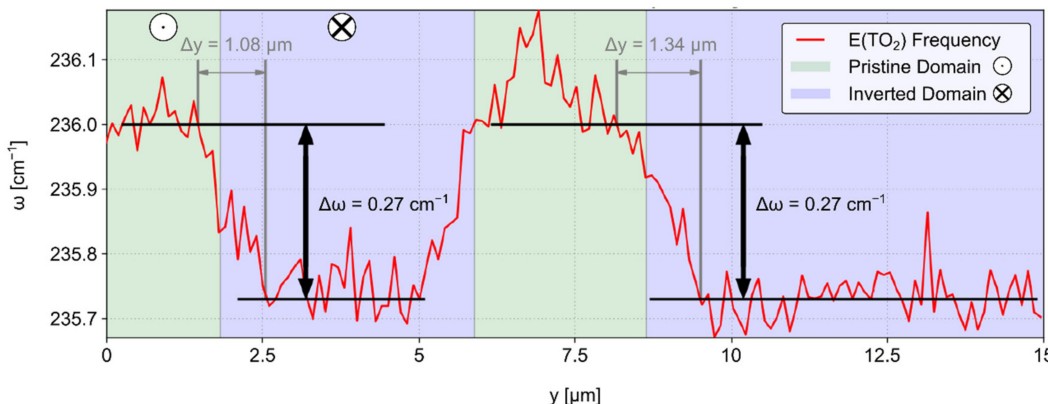

**Figure 11.** Line scan of the $E(TO_2)$ phonon frequency over poled TFLN domains. A sample section with large domains has been chosen for a clear visualization of the phonon frequency shift between pristine and inverted domains, which are highlighted in blue and green as a guide to the eye. The analyzed DW interaction length $\Delta y$ is larger than the resolution limit, suggesting long-range stress fields in the vicinity of the DW.

The presented analysis shows the high versatility of Raman microscopy on TFLN. This method not only provides the domain polarity and domain wall contrast for the inspection of poling period and duty cycle, but is capable of detecting buried DWs and thus probing the in-depth poling completeness. Additionally, stress and electric fields are also potentially accessible. Regarding the multitude of possible influence factors on the detected signals, the challenge is to unequivocally assign the observed features to certain material properties. The cross-correlation with other methods is a viable way to achieve this goal, combined with extended fundamental investigations of the specific property sensitivities. The resolution, similar to SHG imaging, is diffraction limited. However, this resolution already allows experimental statements, as we observe signatures for strain fields that spread significantly further than the optical resolution. Similar observations have been made in bulk in the past as well, e.g., fields of shear strain reaching 10 µm from the DW have been observed [52].

The biggest challenge for the application of RS to TFLN is the low scattering volume of the thin film. The necessary long acquisition times limit the pixel number and applications in the process control. Potentials to improve the signal and hence application times, can be a higher pump power, lower spectral resolution or improved focusing. However, even in bulk, typical acquisition times in the order of 1 s are typical and may be the limit. Raman spectroscopy is thus mainly suited for fundamental investigations of sample properties and the validation of other methods via cross-correlation.

Alongside fundamental investigations that facilitate the quantification of electric and stress fields, angular dispersion experiments on TFLN can give insights on whether the phonon properties are further affected by the boundaries of the thin film. DFT calculations can also help theoretically understand the differences to bulk LN systems [58,60].

In summary, the application of Raman spectroscopy provides imaging with DW and polarity contrast, a sensitivity towards the completeness of in-depth poling, and access to apparent strain and electrical fields. Due to its long acquisition time of several seconds per pixel, the number of data points is considerably lower compared to PFM and SHG, and the application is limited to fundamental investigations. For a better understanding and quantification of the obtained results, further experiments are suggested.

## 4. Conclusions

In this work, we have performed imaging of PP-TFLN devices with PFM, SHG microscopy, and Raman spectroscopy. Moreover, we have discussed the underlying contrast mechanisms and the differences to imaging in related bulk structures. While each method addresses a different physical property to reveal the domain structure, i.e., piezoelectric tensor, the SH tensor, and the phonon-properties, the resulting images are comparable. Hence, each method is capable of visualizing and analyzing the domain structures. Furthermore, each method gives access to additional properties of the domain structure.

Table 1 summarizes the imaging contrasts, resolution, and speed of all the discussed methods. Similar to its use in bulk PPLN, selective etching allows fast imaging of z-cut TFLN. However, due to the additional need to expose z-faces, its speed and information content is limited for x-cut TFLN. If a waveguide is already fabricated, in situ transmission investigations provide almost instant feedback during fabrication. However, since only integral information is retrieved, little quantitative information on the domain structure can be gained. SHG is a promising candidate for standard imaging on TFLN. It offers access to all the determined key performance indicators, i.e., poling period, duty cycle, poling grid length, and poling depth. In the case of submicron poled TFLN structures, however, the high resolution of PFM scans is vital for the acquisition. PFM potentially offers access to most key performance indicators, as well. For fundamental optimization, the sensitivity of Raman scattering for defects, mechanical stress, and/or electric fields gives additional insights into the material system which are not approachable by other techniques. Due to its simultaneous sensitivity to domain structures, it allows studying the interactions and dependencies of domain structures and defects or fields.

**Table 1.** Comparison of the techniques applied to the analysis of x-cut TFLN, with respect to the following parameters: Detectable imaging contrasts, contrast mechanisms, spatial resolution, measurement speed, acquisition time, destructive nature of the technique. PFM: Piezoresponse force microscopy; SHG: Second-harmonic-generation microscopy: RS: Raman micro-spectroscopy.

| | **In Situ** | **Etching** | **PFM** | **SHG** | **Raman** |
|---|---|---|---|---|---|
| **Contrast mechanism** | device performance | etch rate | electrooptic tensor | $\chi^{(2)}$ | phonon properties |
| **DW contrast** | - | x | x | - | x |
| **Polarity contrast** | - | - | x | x | x |
| **In-depth poling** | - | - | (x) | x | (x) |
| **Strain & el. fields** | - | - | - | - | x |
| **Resolution** | - | 50–100 nm | 1–100 nm | 0.5–1 $\mu$m | 0.5–1 $\mu$m |
| **Typical frame size** | no imaging | $\mu$m$^2$ to mm$^2$ (z-cut) 1–20 $\mu$m$^2$ (x-cut) | 100–1000 $\mu$m$^2$ | mm$^2$ | 100 $\mu$m$^2$ |
| **Time per frame** | In situ | - | ~1–10 min | 1 s–10 min | 1 d–1 week |
| **Destructivity** | no | yes | no | no | no |

For future developments, the combination of these imaging methods paves the way for highly efficient TFLN platforms. However, a complete understanding of the underlying imaging mechanisms and quantification of signals is necessary for further TFLN optimization. For PFM, the sensitivity for in-depth poling needs to be validated. In SHG measurements, the depth sensitivity of the signal must be further quantified. For RS, the quantification of the stress sensitivity and further understanding of the different imaging mechanisms will provide deeper insights into the nature of poled TFLN samples.

With these future investigations, the imaging and in-depth signal analysis of TFLN structures provide a valuable toolkit for the development of nonlinear optical devices with maximum conversion efficiency.

**Supplementary Materials:** The following are available online at https://www.mdpi.com/2073-435 2/11/3/288/s1. Figure S1: Colormaps of $A_1$(TO) frequencies, intensities, and peak widths.

**Author Contributions:** Conceptualization, S.R., M.R., and L.M.E.; methodology, S.R. and M.R.; formal analysis, S.R. and M.R.; investigation, S.R., B.K., and M.R.; resources, J.Z., S.M., and L.M.E.; writing—original draft preparation, S.R. and M.R.; writing—review and editing, M.R. and L.M.E.; visualization, S.R.; supervision, L.M.E.; project administration, L.M.E.; funding acquisition, S.M. and L.M.E. All authors have read and agreed to the published version of the manuscript.

**Funding:** This research was funded by Deutsche Forschungsgemeinschaft (DFG) via FOR5044, EN 434/41-1 and INST 269/656-1 FUGG, Sandia National Laboratories (SigmaNONlin), and the National Science Foundation (EFMA-1640968).

**Institutional Review Board Statement:** Not applicable.

**Informed Consent Statement:** Not applicable.

**Acknowledgments:** The authors would like to thank the Light Microscopy Facility of the CMCB Technology Platform at TU Dresden for the use of the SHG microscope Leica SP5 MP. Open Access Funding by the Publication Fund of the TU Dresden.

**Conflicts of Interest:** The authors declare no conflict of interest.

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
