# Peer review of "“Seeing Is Believing”—In-Depth Analysis by Co-Imaging of Periodically-Poled X-Cut Lithium Niobate Thin Films"

_crystals, doi:10.3390/cryst11030288_

Round 1

Reviewer 1 Report

The authors present a lengthy and well written work that is essentially an experimental review of techniques used to characterise periodic poling in lithium niobate thin films. They conduct original experiments using previously published techniques to compare the strengths and weaknesses of these measurement approaches. I believe it is of broad interest as a topical review of this subject.

My only comment relating to the content is as follows

In Figure 8 can the authors discuss why the measured intensity changes in surrounding regions that are not poled?

Author Response

Please see page 2 of the attachment.

Reviewer 2 Report

Comments to the Author

In the manuscript, the authors discuss three different analyses of periodically-pole lithium niobate thin films. The article is interesting and very well written. All techniques are appropriately considered. Figure 3 and Table 1 are very interesting and instructive. I recommend the paper for publication.

Minor suggestions (not obligatory, authors can decide):

  • Page 2 - The key indicators in lines 52, 53 “(a) the appropriate poling period; (b) the 53 duty cycle; (c) the poling length; and (d) the depth of poled domains” are different then the key indicators described below on the same page: “a) Sub-micrometre resolution, b) Duty-cycle and homogeneity, c) Speed and d) In-depth domain growth control. Please unify them.
  • Authors could include also the latest reviews on these techniques, such as http://dx.doi.org/10.1098/rspa.2018.0782.
  • Unify the addressing of images in the text: Figure or Fig.

Author Response

Please see page 3 of the attachment.
